# The Therapeutic Potential of Secreted Factors from Dental Pulp Stem Cells for Various Diseases

**DOI:** 10.3390/biomedicines10051049

**Published:** 2022-05-02

**Authors:** Kenichi Ogata, Masafumi Moriyama, Mayu Matsumura-Kawashima, Tatsuya Kawado, Aiko Yano, Seiji Nakamura

**Affiliations:** Section of Oral and Maxillofacial Oncology, Division of Maxillofacial Diagnostic and Surgical Sciences, Faculty of Dental Science, Kyushu University, 3-1-1 Maidashi, Higashi-ku, Fukuoka 812-8582, Japan; moriyama@dent.kyushu-u.ac.jp (M.M.); mayu8mayu@dent.kyushu-u.ac.jp (M.M.-K.); t.kawado@dent.kyushu-u.ac.jp (T.K.); a.yano@dent.kyushu-u.ac.jp (A.Y.); seiji@dent.kyushu-u.ac.jp (S.N.)

**Keywords:** dental pulp stem cells, secreted factors, anti-inflammatory, immunosuppression, M2-type macrophages

## Abstract

An alternative source of mesenchymal stem cells has recently been discovered: dental pulp stem cells (DPSCs), including deciduous teeth, which can thus comprise potential tools for regenerative medicine. DPSCs derive from the neural crest and are normally implicated in dentin homeostasis. The clinical application of mesenchymal stem cells (MSCs) involving DPSCs contains various limitations, such as high cost, low safety, and cell handling issues, as well as invasive sample collection procedures. Although MSCs implantation offers favorable outcomes on specific diseases, implanted MSCs cannot survive for a long period. It is thus considered that their mediated mechanism of action involves paracrine effects. It has been recently reported that secreted molecules in DPSCs-conditioned media (DPSC-CM) contain various trophic factors and cytokines and that DPSC-CM are effective in models of various diseases. In the current study, we focus on the characteristics of DPSC-CM and their therapeutic potential against various disorders.

## 1. Background

The oral environment, which consists of teeth, gums, and the tongue, is one of the most specialized formations of the craniofacial skeleton and forms an excellent developmental model of organ formation. In addition, this complex and sophisticated structure offers great potential for regenerative medicine due to the abundant resources of highly specialized mesenchymal stem cells (MSCs). The most recently discovered stem cells in the mesenchymal stem cell repertoire, dental stem cells, appear to have immense potential for developmental, differentiation, regeneration, and immunomodulatory/immunomodulatory properties. The discovery of dental stem cells has made great strides toward research and clinical application, highlighting the importance of this interesting source of stem cells. 

Dental pulp stem cells (DPSCs) deriving from child or adult human dental pulp were initially discovered by Gronthos and colleagues in 2000 [1], although the presence of stem cells in dental pulp has been previously reported by Yamamura in 1985 [2]. DPSCs were isolated on the basis of high proliferation and colonization frequency, producing sporadic but densely calcified nodules. In addition, DPSCs do not significantly differ from MSCs deriving from adipose tissue, bone marrow, and umbilical cord tissue with respect to their internationally accepted vague criterium of plastic adherence and their ability to differentiate into osteoblasts, chondrocytes, and adipocytes in vitro while expressing mesenchymal (CD29, CD90, CD105, CD73, and CD44) but not hematopoietic lineage markers (CD14, CD34, and CD45) [3]. It was suggested by the Committee of Mesenchymal Stem Cells and Tissues of the International Society for Cellular Therapy in 2005 that 95% of human MSCs expresses at least the following surface antigens, CD105 (endoglin), CD73 (5′-ectonucleotidase), and CD90/Thy-1 (glycosylphosphatidylinositol-anchored glycoprotein); however, they do not express (less than 2%) the surface antigens CD11b, CD14, CD19, CD34, CD45, CD79a, and HLA-DR. Nevertheless, other studies have reported different cell surface antigens in the prospective isolation of MSCs, such as STRO-1 (stromal precursor antigen 1), VCAM-1 (vascular cell adhesion molecule 1), SH2 (Src homology 2), SH3/SH4, CD271, GD2 (ganglioside 2), and SSEA4 (stage-specific embryonic antigen-4) [4,5,6,7,8]. Thus, there are various differences in the expression of surface markers of DPSCs depending on the reports.

MSCs are known to be involved in growth, wound healing, and cell replacement under both normal and disease states. These cells have demonstrated effectiveness in regenerating periodontal tissue, diabetic critical limb ischemic tissue, bone damage triggered by osteonecrosis, skin lesions after burns [9,10,11], as well as liver, neuronal and skeletal muscle tissues, and blood vessels [12,13,14,15] among other tissues [16]. To date, there have been many preclinical experiments using DPSCs, and there are reports that DPSCs are differentiated into various cells, e.g., neural cells, dopaminergic neurons, myotubes, hepatocyte-like cells, and keratocytes [17,18,19,20,21]. However, there are no reports on large-scale clinical trials. The problem with using stem cells is the risk of cancer and the necessity for special cell processing facilities.

Various previous studies have demonstrated that transplanted cells can have multiple and significant roles; they not only can travel in their host tissues and participate directly in tissue regeneration [22], they can also present paracrine effects [23,24,25,26,27]. DPSCs produce cytokines that can reduce inflammation, increase progenitor cell proliferation, ameliorate tissue repair, and reduce infection more effectively compared to bone marrow-derived MSCs (BMMSCs) [28]. The goal of this review is thus to highlight the therapeutic effect of secreted factors from DPSCs for various diseases as well as their mechanism of action.

## 2. Cytokines Contained in DPSC-CM

As presented in Table 1, previous reports have shown that DPSC-conditioned media (DPSC-CM) contain more anti-inflammatory cytokines than BMMSC-conditioned media (BMMSC-CM) [29,30,31], e.g., interleukin (IL)-10, 34 times higher; IL-13, 63 times higher; follistatin, 15 times higher; transforming growth factor (TGF)-β1, 7 times higher (Table 1). Moreover, hepatocyte growth factor (HGF), neural cell adhesion molecule-1 (NCAM-1), adiponectin, ectodomain of sialic acid-binding Ig-like lectin-9 (ED-Siglec-9), matrix metalloproteinase (MMP)-13, neurotrophin-3 (NT-3), brain-derived neurotrophic factor (BDNF), and MMP-9 were detected only in DPSC-CM. Based on the above, DPSC-CM contain a higher number of anti-inflammatory cytokines compared to BMMSC-CM.

## 3. Cerebrovascular Disease

There are some studies regarding conditioned media therapy of the central nervous system (CNS) [32,33,34,35]. Engraftment of DPSCs promotes the functional recovery from various acute and chronic CNS insults through paracrine mechanisms that induce endogenous tissue-repairing activities [36,37,38,39]. The mechanisms of DPSC-CM for CNS are summarized below.

Stroke is the third leading cause of death globally and the most common cause of long-term disability in humans [40]. Lately, transplantation of bone marrow mononuclear cells was revealed to accomplish clinical efficacy by promoting angiogenesis in patients with cerebral ischemia [40,41]. In a focal cerebral ischemia animal model, DPSC-CM and BMMSC-CM equally inhibited the expression of pro-inflammatory cytokines and markers of oxidative–nitrosative stress [32]. Conversely, DPSC-CM activated M2-type microglia selectively, leading to the expression of the mRNA-encoding BDNF, a neurotrophin that has a key role in the synaptic remodeling correlated with memory development in the adult hippocampus [32]. In an aneurysmal subarachnoid hemorrhage (aSAH) rat model, a potential synergistic effect from DPSC-CM was observed. In particular, high levels of tissue inhibitor of metalloproteinases (TIMP)-2 in DPSC-CM were observed, which are crucial for the maintenance of tissue homeostasis [35]. TIMP-2 is an inhibitor of MMP-3 and -9 and has been shown to be significantly associated with an increased risk of soft tissue trauma, such as Achilles tendinopathy [42]. With respect to the functional study of TIMP-2 in the neural system, it was considered to be substantially valuable for noise-related cochlear injury [43].

## 4. Heart Disease

Ischemic heart diseases, as for instance myocardial infarction (MI), are among the leading causes of death worldwide [44]. MI usually results in irreversible damage in heart tissues, and the associated mortality and morbidity remain high [45]. The development of effective adjunctive therapies for patients with MI is thus still essential.

Administration of DPSC-CM reduced MI and enhanced cardiac function in mice after ischemia reperfusion (I/R). This correlated with ischemic heart apoptosis and reduced inflammation [46,47]. Yamaguchi S et al. showed that DPSC-CM induced cardiomyocyte survival in response to hypoxia and serum deprivation, and that DPSC-CM reduced the expression of pro-inflammatory mediators promoted by lipopolysaccharide (LPS) [46]. DPSC-CM administration can thus protect the heart from ischemic damage by at least two mechanisms mediating reduction of cardiomyocyte death and suppression of inflammatory responses in myocardial cells.

## 5. Lung Disease

Acute respiratory distress syndrome (ARDS) is a devastating inflammatory disease characterized by diffuse pulmonary injury and consequent fibrosis, leading to respiratory failure and death [48,49]. ARDS development takes place in two phases; the exudative phase is characterized by loss of alveolar structure and accumulation of inflammatory cells at the site of injury. The shift from the exudative phase to the proliferative phase is characterized by fibroblast proliferation and differentiation in myofibroblasts and excessive extracellular matrix deposition. Once remodeling was confirmed, the fibrosis evolves to an irreversible state that further accelerates pulmonary dysfunction and raises mortality rates. The development of efficient treatments focused on reversing or suppressing ARDS progression is currently studied. The pro-inflammatory M1 and anti-inflammatory M2-type macrophages are considered to represent the extreme activation states of macrophage activation [50,51,52].

In mouse bleomycin (BLM)-induced acute lung injury (ALI), administration of DPSC-CM enhanced BLM-induced lung fibrosis, at least partially, via the conversion of a pro-inflammatory M1 environment to an anti-inflammatory M2 [53]. In contrast, BMMSC-CM relatively reduced the BLM-induced pro-inflammatory response but did not promote M2 cell differentiation or enhancement of pulmonary fibrosis. These results can be in contrast to these in a previous study in which the administration of mouse BMSC-CM was shown to promote M2 macrophage differentiation in the injured lungs of lipopolysaccharide-treated mice [54]. Moreover, DPSC-CM stimulated the in vitro differentiation of mouse bone marrow-derived macrophages into M2 macrophages expressing Arginase 1, CD206 and Ym-1 in an in vitro study. Thus, treatment with DPSC-CM induces an anti-inflammatory M2 environment and significantly ameliorates the injured lung’s state.

## 6. Liver Disease

Acute liver failure (ALF), which can be caused by various parameters such as drugs, viruses, and ischemiatoxins, can lead to massive hepatocyte destruction and an unlimited inflammatory response [55]. Although the liver has inherent tissue-repairing activities that induce the division of mature hepatocytes and the proliferation or differentiation of adult liver stem/progenitor cells (LPCs) [56], when acute tissue loss overcomes the liver’s regenerative capacity, the impaired hepatic function results in systemic inflammation, multiple organ failure and sudden death [57]. When ALF becomes chronic, it evolves to liver fibrosis (LF), which is a result of many chronic liver diseases, such as viral infection and autoimmune hepatitis, alcohol abuse, and the use of specific drugs [58].

In rat ALF models, BMMSC-CM was revealed to present therapeutic benefits [59,60]. Nevertheless, which effect was limited. Matsushita Y et al. and Hirata M et al. recommended that DPSC-CM comprised various efficient factors for ALF treatment [61,62]. HGF is a multifunctional protein that prevents hepatocyte apoptosis and induces the proliferation of LPCs and neovascularization [63,64]. Stem cell factors (SCF) inhibit hepatic apoptosis in acetaminophen-induced ALF and is involved in the induction of LPCs [59,65]. Insulin-like growth factor-binding protein (IGFBP)-1 and TIMP-1 inhibit hepatocyte apoptosis in ALF promoted by various stimuli [66,67]. Angiogenin, vascular endothelial growth factor (VEGF)-A and endocrine-derived vascular endothelial growth factor (EGVEGF) are established angiogenic promoters that play key roles in neo-vascularization after liver injury [68,69,70]. In addition, MCP-1 and IL-6 induce survival of human CD11b^+^ peripheral blood mononuclear cells and M2-type macrophage polarization [71]. Favorable factors included within the DPSC-CM and endogenous tissue-repairing factors induced by DPSC-CM therapeutic function in combination attenuate D-galactosamine-induced ALF or carbon tetrachloride-induced LF.

## 7. Eye Disease

Retinitis pigmentosa (RP) is a hereditary disorder associated with photoreceptor degeneration and loss. Its inheritance can be autosomal dominant, autosomal recessive, or X-linked. RP has varying incidence in different ethnic groups, and it affects globally 1 in 4000 people, concerning a total of over 1 million individuals [72]. Some RP patients become blind with aging; approximately 25% become officially blind (20/200), and 0.5% become completely blind in both eyes [73]. The discovery of an effective and safe treatment for retinal degenerative diseases would significantly help patients and support reduction of the economic burden on society.

Li XX et al. showed that subretinal injection enables DPSCs or DPSC-CM to contact the photoreceptors directly [74]. It was demonstrated that DPSC-CM resulted in a similar therapeutic effect as DPSCs [74,75]. Although the main effect is believed to be its anti-apoptotic activity, the pathway to the effects of DPSC-CM, which lasts for several months, remains unclear by further studies of eye disease.

## 8. Neurological Disorder

Spinal cord injury (SCI) is commonly caused by an accident or a fall, developing in a disorder in motor or sensory function [76]. When the spinal cord is injured, late effects such as paralysis of the limbs may persist, which may result in severe difficulties in daily life. The development of SCI is significantly associated with the development of inflammation at the site of injury. There are two types of macrophages, M1 and M2. M1 macrophages digest debris and are involved in inflammation, while M2 macrophages act to reduce inflammation. M1 macrophages promote inflammation by secreting pro-inflammatory cytokines and nitric oxide. M1 macrophages also play an important role in the recruitment and activation of astrocytes. Conversely, M2 macrophages induce tissue repair by releasing anti-inflammatory cytokines [77,78,79,80]. Transplantation of DPSCs in a mouse model after SCI presented in previous studies was found to secrete various trophic factors, such as BDNF and GDNF, and significantly enhanced motor function [81,82]. Furthermore, released ED-Siglec-9 and monocyte chemoattractant protein-1 (MCP-1), which are found specifically in DPSC-CM, induced significant functional recovery in a rodent SCI model by promoting an M2-dominant neuro-repairing microenvironment [83]. Additionally, Kano F et al. demonstrated that MCP-1 and ED-Siglec-9 are crucial for DPSC-CM-mediated functional recovery after facial nerve injury (FNI) in a rat model [84]. Importantly, MCP-1/sSiglec-9 promoted polarization of M2 macrophages, which competed the pro-inflammatory M1 conditions correlated with FNI, induced proliferation, migration, and differentiation of Schwann cells, and increased neurite extension of peripheral nerves.

Parkinson’s disease (PD), the second most common neurodegenerative disease, is characterized by gradual loss of dopaminergic neurons in the substantia nigra [85,86]. In a PD animal model, DPSC-CM-mediated mechanisms may involve C α-synuclein accumulation and reduced levels of CD4-positive cells [87]. 

Alzheimer’s disease (AD) is a developing neurodegenerative disease, characterized by deterioration of cognitive function linked with the deposition of β-amyloid (A*β*) peptides in the brain [88,89,90,91]. Nevertheless, the exact pathways involved in Aβ-induced neurotoxicity and neuroinflammation are not currently understood. In an AD-like mouse model, DPSC-CM inhibited 6-hydroxydopamine-induced cell death and promoted the neurite growth of cerebellar granule neurons [92], as DPSC-CM comprised various types of neuroprotective factors including BDNF, HGF, MMP-9, and TGF-β. In particular, DPSC-CM promoted specifically M2-type microglia, leading to expression of the mRNA encoding BDNF, a neurotrophin playing a key role in the synaptic remodeling correlates with memory formation in the adult hippocampus. The multidimensional activities of DPSC-CM may support many neuro-reparative effects required for the therapy of cognitive deficits.

Amyotrophic lateral sclerosis (ALS) is a fatal neurodegenerative disease that causes destruction of upper and lower motor neurons in the CNS [93]. After diagnosis, the disease typically progresses to advanced stages, and there are limited therapeutics and no cure for ALS mainly because of the disease presentation characteristics. Using an ALS mouse model, Wang J et al. assessed the therapeutic effect of DPSC-CM systemic administration [94]. However, the mechanism involved in DPSC-CM’s therapeutic benefit remains unclear. In early disease, systemic DPSC-CM administration can directly cause a therapeutic benefit to spinal neurons or cells involved in neuromuscular junctions, including Schwann and muscle cells. As demonstrated in Table 1, DPSCs secrete various potent neurotrophic factors, such as NGF, BDNF, VEGF, and NT, which have established therapeutic benefits for SCI and other neurodegenerative diseases [35,36,82,95]. These factors could thus directly protect motor neurons, leading to increased survival.

Multiple sclerosis (MS) is a demyelinating disease of the CNS. MS predominantly affects young adults, with an onset average age of approximately 30 years. The origin of MS is not yet clear, but it is considered to develop from immune defects. The main symptoms involve impaired vision, nystagmus, and swallowing and talking difficulties. The protective effect of DPSC-CM has been demonstrated in the treatment of experimental autoimmune encephalomyelitis (EAE), which is one of the MS models. Administration of DPSC-CM decreased expression of inflammatory cytokines in the spinal cord, inhibited demyelination, and enhanced EAE clinical scores [96]. Shimojima C et al. reported that despite the effect of DPSC-CM on EAE depending on ED-Siglec-9, DPSC-CM–treated mice demonstrated relatively improved recovery compared to ED-Siglec-9- treated mice, indicating that other factor(s) in DPSC-CM, including HGF, could cooperatively improve EAE [96].

## 9. Autoimmune Disease

As MSCs are exceptional as immunosuppressive agent candidates during solid-organ transplantation and for treating graft vs. host disease, inflammatory diseases, as well as other autoimmune diseases [97,98]. Early studies have demonstrated that DPSCs block the proliferation of stimulated T cells, with a more robust pattern compared to BMMSCs [99,100]. DPSCs inhibited peripheral blood mononuclear cells (PBMCs) proliferation that was stimulated with mitogen or was in an allogeneic-mixed lymphocyte reaction, suggesting that the immunosuppressive properties of MSC-like cells are expected to include soluble factors mediated by signaling molecules derived from activated PBMCs [101]. In previous studies, it has been demonstrated that toll-like receptors are generally distributed in cells of the immune system and can activate immunosuppression of the DPSCs via improving the expression of TGF-β and IL-6 [102,103].

Most recently, Matsumura-Kawashima M et al. administrated DPSC-CM in non-obese diabetic (NOD) mice as primary Sjögren’s syndrome (SS), which is a chronic, systemic autoimmune condition characterized by inflammation of exocrine glands and functional impairment of the salivary and lacrimal glands [30]. DPSC-CM developed a protective effect on the secretory function of salivary glands (SMGs). Furthermore, DPSC-CM improved hyposalivation caused from SS by reducing inflammatory cytokine expression, promoting regulatory T cells (Treg) in the spleen via the TGF-β/Smad pathway, controlling the local inflammatory microenvironment, and reducing apoptosis in SMGs. Accordingly, Ogata K et al. discovered that DPSC-CM alleviated hyposalivation caused by SS in MRL/MpJ-faslpr/faslpr mice, which can take place in accordance with other rheumatic diseases (secondary SS) models, by reducing the number of inflammatory cytokines, controlling the local inflammatory microenvironment, and reducing apoptosis in SMGs [31].

## 10. Bone and Cartilage Disease

Articular cartilage (AC) plays vital roles in the operation of diarthrodial (synovial) joints [104,105]. Cartilage injuries are relatively common, mainly in young and active athletes, especially in the knee joint [106,107,108]. They are often thought as risk factors for the development of osteoarthritis (OA) in later life, a degenerative and inflammatory disease of the synovial joint with irreversible cartilage loss [104]. OA causes disability, especially in the elderly, and correlates with a large socio-economic burden [109,110]. Matrix-induced autologous chondrocyte implantation (MACI) has been developed aiming to restore damaged cartilage tissue [111]. Nevertheless, MACI contains several limitations such as potential iatrogenic damage and high associated costs [112,113,114]. To overcome these limitations, the use of innovative autologous tissue engineering approaches using stem cells has created an increasingly interesting area aimed at achieving AC regeneration.

DPSC-CM can promote endogenous cells to proliferate and replace the lost tissue, while it can inhibit progression of cartilage loss by impairing chondrocyte apoptosis [115]. Furthermore, Lo Monaco M et al. suggested that DPSC-CM might cause multiple anti-inflammatory and anti-catabolic effects in OA chondrocytes [115]. Elucidation of the paracrine effects of DPSCs and improved understanding of stem cell regulation provide researchers with a significant number of trauma treatment options that have been limited by cell procurement concerns.

The temporomandibular joint (TMJ) is a load-bearing articulator with various motions comprising rotation and translation [116]. The AC in a typical synovial joint is a thin layer of hyaline cartilage that covers the entire articular surface of each bone. TMJ osteoarthritis (TMJOA) is a degenerative joint disease, characterized by progressive cartilage degeneration, abnormal bone remodeling, and chronic pain [117]. The mechanisms causing TMJOA are complex and multidimensional. Nevertheless, it has been indicated that increased malocclusion causes mechanical stress impacts during TMJOA development [118]. In the advanced stage, the tissue-degenerated TMJOA microenvironment induces cartilage matrix degradation, chondrocyte apoptosis, and necrosis; In addition, irregular resorption of subchondral bone causes irreversible joint damage and dysfunction [119,120].

Ogasawara N et al. employed a mechanical-stress-associated TMJOA model comprising five consecutive days of TMJ damage followed by treatment with DPSC-CM for the following five days after TMJ damage [121]. DPSC-CM comprised various therapeutic factors with the ability of TMJOA treatment. DPSC-CM not only prevented the articular degradation cascade but also regenerated the injured articular joint by inducing proliferation of the multipotent polymorphic cell layer and development of the cartilage matrix in mice with TMJOA. Furthermore, they discovered several exosome markers such as Alix, tumor susceptibility 101 (TSG101), integrin α chains, and tetraspanin (TSPANs, CD9, CD63, CD81) that are representative members of a protein superfamily characterized by the occurrence of four transmembrane and two extracellular domains. This indicates that DPSCs may secrete a higher number of exosomes compared to what was previously considered [121,122]. Nevertheless, in order to elucidate the exact mechanisms of DPSC-CM-mediated AC regeneration, the evaluation of therapeutic factor accessibility in DPSC-CM toward injured TMJ is essential.

Bisphosphonate (BP) therapeutics are valuable in osteoporosis treatment, Paget disease of bone, bone metastasis from malignant tumors, multiple myeloma, and hypercalcemia, as well as in inhibiting and improving a range of bone-associated conditions [123,124]. Nevertheless, BP preparations tend to promote osteoclast (OC) apoptosis and inhibit angiogenesis [125,126]. Despite the fact that drug-related osteonecrosis of the jaw (MRONJ) has been described as a symptom associated with BP preparations, no effective treatment has yet been discovered. Several therapeutic approaches have been suggested to date aiming to increase conservative and non-invasive modalities [127,128], but these methods provide a limited delay on symptom progression and fail to accomplish complete remission. Various reports involved DPSC-CM administration on a MRONJ rat model generated using sodium zoledronate (ZOL) administration, and after performing dental extraction, they evaluated the effects of DPSC-CM on OC and osteoblasts (OB), and assessed osteonecrosis and angiogenesis improvement [129,130]. DPSC-CM comprises angiogenic growth factors, such as VEGF (Table 1) and IL-8 [131]. These factors may promote epithelial tissue formation by increasing vascular endothelial cell numbers and by inducing chemotaxis [131], thus improving bone exposure. DPSC-CM were also found to express MCP-1 (Table 1). Based on previous reports, OC precursors result from monocytes. MCP-1 is a monocytic, chemotactic protein that induces monocyte and lymphocyte chemotaxis and increases IL-1 and IL-6 secretion [132,133]. The protein may promote OC synthesis by increasing the accumulation and induction of OC precursors.

## 11. Metabolic Disease

Diabetes mellitus is a metabolic condition characterized predominantly by chronic hyperglycemia, which is caused from insulin resistance or from impairment of insulin secretion. In total, 75% of obese patients do not develop diabetes due to the effect of compensatory insulin secretion; maintenance of functional pancreatic β-cells is thought to be the ultimate result of diabetes treatment [134]. Existing antidiabetic drugs are inadequate to reduce progressive damage in pancreatic β-cells. As a result, patients need to eventually switch to insulin therapy [135]. Islet transplantation and regenerative medicine thus continue to receive increased interest as a developing treatment option for diabetes [136]. Moreover, diabetic polyneuropathy (DNP), the most common diabetic complication in both type 1 and type 2 diabetes, impacts up to 50% of these patients [137]. Symptoms of DPN include spontaneous pain, hyperalgesia, and diminished sensation [138]. Pathogenesis of DPN involves degeneration of nerve fibers and decreased nerve blood flow. Tight glycemic control is essential for the reduction of DPN progression but not its complete inhibition [139]. Additionally, multifactorial intervention targeting hyperglycemia, hypertension, and dyslipidemia failed to decrease the risk of DPN in patients with type 2 diabetes [140]. Although symptomatic therapy drugs for painful DPN are useful for amelioration of patients’ quality of life, essential therapies for the pathogenesis of DPN are still necessary.

DPSC-CM regenerated pancreatic β-cells in streptozotocin (STZ)-stimulated diabetic mice, in which pancreatic β-cells were substantially destroyed in this mouse. Izumoto-Akita T et al. recommended that factors secreted from stem cells (paracrine signals) could be responsible for this process [141]. Previous studies have demonstrated that VEGF plays a crucial role in the proliferation and development of pancreatic β-cells [142], and stimulation of phosphoinositide 3-kinase suppresses cell death, proliferation, and pancreatic β-cells function [143]. Moreover, examination of fractionated DPSC-CM based on molecular weight revealed that compared to DMEM, a fraction of more than 100 kDa enhanced viability of STZ-treated MIN6 cells (mouse pancreatic β-cell line) [141]. DPSCs have been described to differentiate into pancreatic β-like cells [144], indicating that DPSC-CM commonly comprise secreted factors affecting differentiation pathways, function, and secretion among neural cells and pancreatic β-cells. Based on the above, DPSC-CM therapy was more efficient in stimulating growth and insulin secretion in pancreatic β-cells compared to BMMSC-CM therapy.

With respect to DNP, the combination of angiogenesis, neuronutrition, and immunomodulatory results could indicate multiple improvements in diabetic polyneuropathy. VEGF gene transfer to the lower limbs enhanced vascularity as well as the symptoms of diabetic polyneuropathy [145,146]. Based on that, the therapeutic mechanisms of DPSC-CM on DNP are the following: (1) angiogenesis and blood flow improvement, (2) anti-inflammatory effect, and (3) neuroprotective effect [147,148,149].

## 12. Dental Diseases

Dental trauma, restorative operative procedures, and/or caries lesions may expose the dental pulp [150]. Coping with this clinical condition in dentistry is challenging, as maintenance of the dentin–pulp complex vitality is crucial. When the exposure of dental pulp takes place in immature teeth with no complete root formation, this problem is even worst since the rhizogenesis procedure is disrupted [151].

Achieving ischemia or necrosis during enhanced tissue cell-based regeneration depends on functional re/angiogenesis and functional survival of transplanted cells [152]. The ability of these cells to regenerate is based on their proliferation, integration, and differentiation characteristics, and on the potential of secretory proteins, such as trophic factors, growth factors, cytokines, and chemokines, which positively affect damaged tissue [153]. These secreted factors, when used without the stem cells, can trigger tissue repair in various forms of organs and tissues damage [154].

The rats’ dental canals treated with DPSC-CM showed the formation of connective tissue, allowing for the detection of scattered blood vessels. Most of them were covered with blood cells, inflammatory cells, and collagen fibers. These data revealed that DPSC-CM were able to perform slight root restructuring of the dental canal. An immunopositively reaction against VEGF receptor 2 has been reported in newly formed tissues associated with the developmental stage of blood vessels [155]. Recently, a study demonstrated that association of DPSC-CM and mineral trioxide aggregate (MTA) revealed favorable effects on direct pulp capping in rats, greater than those of MTA when applied alone. Importantly, improvements were observed in the rate of dentin bridge formation, the morphological quality of these newly formed hard tissues, and primarily the control of inflammation in the pulp tissue [156].

## 13. Relationship between DPSC-CM and Exosomes

Primarily, the DPSC-CM comprises cytokines and a miniscule number of DPSC-secreted exosomes, which is easier to obtain in cytokines (Table 1). Hence, DPSC-CM is assumed to be nonimmunogenic, promoting their use in the same or different animal species. The action mechanism of the secreted factors mainly regulates the balance between these anti- and pro-inflammatory cytokines in a specific tissue (Table 2).

Exosome refers to a specific class of lipid-membrane-bound extracellular vesicles having a diameter of 40–150 nm. Subsequently, exosomes may be internalized by other cells, mainly by phagocytosis and fusion with the cell membrane and receptor–ligand interaction, allowing them to release their contents into the cytoplasm [157]. The exosomes’ regenerative potential might be modulated by various mechanisms, including the prior exposure of the originating cell population to external stimuli [158]. However, the primary functional molecules of exosomes remain unclear, and currently, there is no unified dosage, method of administration, or storage scheme for cell-free therapy; thus, further systematic and extensive research is required to resolve these problems.

## 14. Current Limitation and Perspective

Since the CM also contains several waste products produced by cell metabolism, there are limits to its therapeutic application [159]. Additionally, there is a difference in the effect of the obtained CM depending on the condition of the cell culture. To apply the CM to clinical practice, it is necessary to mass-produce it using an automatic culture device.

## 15. Conclusions

Figure 1 and Table 2 summarize the relevant reports. DPSC-CM may be more relevant for cell-free treatment of immune- and inflammation-related diseases. Nevertheless, studies evaluating the immunomodulatory and anti-inflammatory effects of DPSC-CM are still insufficient compared with BMMSC-CM. Additional studies should focus on the full elucidation of the critical role and the associated mechanisms of DPSC-CM immunomodulatory properties.

## Figures and Tables

**Figure 1 biomedicines-10-01049-f001:**
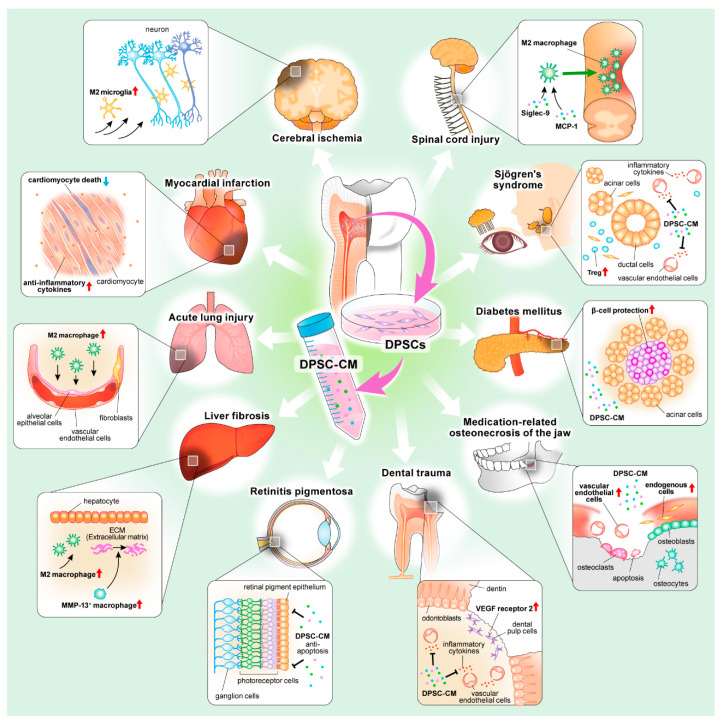
Schematic representation of therapeutic potential summary of DPSC-CM.

**Table 1 biomedicines-10-01049-t001:** The anti-inflammatory factors or DPSC-CM only detected factors of DPSC-CM vs. BMMSC-CM.

	**Anti-Inflammatory Factors** **(Intensity)**	
	**BMMSC-CM**	**DPSC-CM**
Follistatin	2328	36241
TGF-β1	2186	11623
IL-10	234	7989
IL-13	80	5098
VEGF	1242	3549
IGF-1	3324	3521
HGF	0	1857
TECK	1513	1609
MCP-1	483	965
IL-29	1019	943
Adiponectin	0	502
ED-Siglec-9	0	396
GM-CSF	188	159
	**Secreted Factors Only in DPSC-CM** **(Intensity)**	
	**BMMSC-CM**	**DPSC-CM**
HGF	0	1857
NCAM-1	0	573
Adiponectin	0	502
ED-Siglec-9	0	396
MMP-13	0	377
NT-3	0	249
BDNF	0	179
MMP-9	0	136

Abbreviations: TGF-β1: transforming growth factor-β1; IL-10: interleukin-10; IL-13: interleukin-13; VEGF: vascular endothelial growth factor; IGF-1: insulin-like growth factor-1; HGF; hepatocyte growth factor; TECK: thymus-expressed chemokine; MCP-1: monocyte chemoattractant protein-1; IL-29: interleukin-29; ED-Siglec-9: ectodomain of sialic acid-binding immunoglobulin-type lectin-9; GM-CSF: granulocyte macrophage colony-stimulating factor; NT-3: neurotrophin-3; BDNF: brain-derived neurotrophic factor.

**Table 2 biomedicines-10-01049-t002:** Therapeutic effects of DPSC-CM for various diseases.

Type	Disease	Animal Model	Administration Method	Mechanism	Reference
Cerebrovascular disease	Forcal cerebral ischemia	Cerebral ischemia model (Sprague-Dawley rats)	Intranasal injection	Induction of activated M2-type microglia	Inoue T et al. [32]
	Aneurysmal subarachnoid hemorrhage	Experimental aSAH model (Wistar rats)	Intrathecal injection	Inhibitor for MMP-3 and 9 by TIMP-2	Chen TF et al. [35]
Heart disease	Myocardial infarction	Myocardial ischemia-reperfusion injury model (C57BL/6J mice)	Intravenous injection	Reduction of cardiomyocyte death and suppression of inflammatory responses	Yamaguchi S et al. [46]
Lung disease	Acute lung injury	Bleomycin-induced acute lung injury model (C57BL/6J mice)	Intravenous injection	Induction of M2 macrophage differentiation	Wakayama H et al. [53]
Liver disease	Liver fibrosis	Carbon tetrachloride (CCl_4_)-induced liver fbrosis model (C57BL/6J mice)	Intravenous injection	Induction of the MMP-13^+^ restorative hepatic macrophages	Hirata M et al. [61]
	Acute liver failure	D-galactosamine-induced acute liver failure model (Sprague-Dawley rats)	Intravenous injection	Induction of an anti-inflammatory M2 environment and activation of adult LPCs	Matsushita Y et al. [62]
Eye disease	Retinitis pigmentosa	C57BL/6 J mice with RPGR knockout	Subretinal injection	Anti-apoptotic activity	Li XX et al. [74]
Neurological disorder	Spinal cord injury	Contusion model of spinal cord injury (Sprague-Dawley rats)	Injection into the injury epicenter	Induction of an M2-dominant neurorepairing microenvironment by ED-Siglec-9 and MCP-1	Asadi-Golshan R et al. [82] Matsubara K et al. [83]
	Facial nerve injury	Facial nerves transection model (Sprague-Dawley rats)	Local implantation with an atelocollagen sponge	Induction of the polarization of M2 macrophages by MCP-1/Siglec-9	Kano F et al. [84]
	Parkinson’s disease	Rotenone-induced Parkinson’s disease model (Lewis rats)	Intravenous injection	Upregulation of tyrosine hydroxylase expression	Chen YR et al. [87]
	Alzheimer’s disease	Aβ_1-40_ peptide-induced Alzheimer’s disease model (imprinting control region mice)	Intranasal injection	Activated M2-type microglia	Mita T et al. [92]
	Amyotrophic lateral sclerosis	Transgenic mice (B6SJL-Tg (SOD1G93A)1 Gur/J) expressing the humansuperoxide dismutase 1 (mSOD1^G93A^) mutation	Intraperitoneal injection	Directly protection of motor neurons	Wang J et al. [94]
	Multiple sclerosis	Experimental autoimmune encephalomyelitis (EAE) model (C57BL/6J mice)	Intravenous injection	Induction of M2 macrophage polarization through ED-Siglec-9 by interacting with the sialic acid-bound CCR2	Shimojima C et al. [96]
Autoimmune disease	Sjögren’s syndrome	Nonobese diabetic (NOD) mice	Intravenous injection	Induction of Tregs throug the TGF-β/Smad pathway	Matsumura-Kawashima M et al. [30] Ogata K et al. [31]
Bone and Cartilage disease	Osteoarthritis	Mouse mechanical stress-induced osteoarthritis model (Institute of Cancer Research mice)	Local injection	Stimulation of endogenous cells to proliferate and replace the lost tissue	Lo Monaco M et al. [115] Ogasawara N et al. [121]
	Medication-related osteonecrosis of the jaw (MRONJ)	Zoledronate-induced MRONJ model (Wistar rats)	Local injection	Induced epithelial tissue formation by increasing vascular endothelial cells	Abe F et al. [129] Kushiro H et al. [130]
Metabolic disease	Diabetes mellitus	Streptozotocin-induced diabetic mice (C57BL/6J mice)	Intravenous injection	Protection and encouragment of the propagation for β-cells	Izumoto-Akita T et al. [141]
	Diabetic polyneuropathy	Streptozotocin-induced diabetic rats (Sprague-Dawley rats) Streptozotocin-induced diabetic mice (C57BL/6J mice)	Local injection into the hindlimb skeletal muscles	Angiogenesis and improvement of blood flow, anti-inflammatory action, and neuroprotective action	Makino E et al. [147] Kanada S et al. [148] Hata M et al. [149]
Dental disease	Dental trauma	Orthotopic transplantation model (Wistar rats)	Local injection with a collagen	Immunopositivity reaction for VEGF receptor 2, and mainly controlling pulp tissue inflammation	de Cara S et al. [155] Sarra G et al. [156]

## Data Availability

Not applicable.

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
