# Peer review of "The Therapeutic Potential of Secreted Factors from Dental Pulp Stem Cells for Various Diseases"

_biomedicines, 2022, doi:10.3390/biomedicines10051049_

Round 1
Reviewer 1 Report
- Figure 1 represents the results of an original paper by Ogata K at all, cited in reference number 27, “Dental pulp-derived stem cell-conditioned media attenuates secondary Sjogren's syndrome via suppression of inflammatory cytokines in the submandibular glands. DOI:10.1016/j.reth.2021.01.006. “. This result is very interesting, but it is not appropriate to use the original results, even if originated by the same authors of this review. I suggest to describe in a schematic or illustrative way the anti-inflammatory factors of DPSC-CM without using the figure and table of the above-mentioned paper. For example, you can make a summary table of the descriptive text of figure 1 (from line 83 to 93).
- In each paragraph of your review authors accurately describe the pathology of interest. For the purposes of the review, I suggest to give less prominence to the pathology and more emphasis on therapeutic aspect of dental pulp stem cells factors secreted. You can therefore streamline the introductory text of paragraphs 3 to 12 while maintaining the pathologies treated. In particular, paragraph 8. “Neurological disorder” is too long compared to the other paragraphs.
- I appreciated the use of very dated bibliographic references that represent strong points of scientific literature, but to make your work more readable I suggest to using references as recent as possible. e.g. Dental pulp stem cells and their potential roles in central nervous system regeneration and repair PMID: 23996516 DOI: 10.1002 / jnr.23250, Oral Plaque from Type 2 Diabetic Patients Reduces the Clonogenic Capacity of Dental Pulp-Derived Mesenchymal Stem Cells PMID: 30755774 PMCID: PMC6348930 DOI: 10.1155 / 2019/1516746, Mapping the Secretome of Dental Pulp Stem Cells Under Variable Microenvironmental Conditions PMID: 34553309 DOI: 1007/s12015-021-10255-2, Therapeutic Potential of Dental Pulp Stem Cell Secretome for Alzheimer's Disease Treatment: An In Vitro Study PMID:27403169 PMCID: PMC4923581 DOI: 10.1155/2016/8102478.
- Your conclusions are summarized in attractive figure 2 and new scientific evidence is needed to validate the actual therapeutic potential of factors secreted by dental pulp stem cells. However, it is not clear what the future prospects and directions for the use of this therapeutic approach may be, in your opinion of experts.
Author Response
We will send the reply to the reviewer as an attachment.
Thank you for your help.
Kenichi Ogata

Reviewer 2 Report
The idea of the manuscript is important since the use of dental pulp cells is emerging. However, certain things should be corrected to warrant publication.
- The inclusion of results in a review paper with inadequate methods is not acceptable. Please remove all original results which need to be included in a complete research paper with full methods and justification
- lines 239-250: please remove the underlying
- Please comment on the limitations of dental pulp stem cell use
- section 10 is about bone disease and you describe articular cartilage. Cartilage is different to bone. Please correct.
- Please provide an overview on evidence for the differentiation potential towards different lineages.
Author Response

(The authors gave the same response as above.)

Round 2
Reviewer 1 Report
None